# Research on Blockchain-Enabled Smart Grid for Anti-Theft Electricity Securing Peer-to-Peer Transactions in Modern Grids

**DOI:** 10.3390/s24051668

**Published:** 2024-03-04

**Authors:** Jalalud Din, Hongsheng Su, Sajad Ali, Muhammad Salman

**Affiliations:** 1School of Automation and Electrical Engineering, Lanzhou Jiaotong University, Lanzhou 730070, China; jalalpak@yahoo.com (J.D.); sajad8371@gmail.com (S.A.); 2School of Electrical Engineering, China University of Mining and Technology, Xuzhou 221000, China; salmankhan5780@gmail.com

**Keywords:** energy theft detection, block chain, smart grids, privacy, P2P computing

## Abstract

Electricity theft presents a significant financial burden to utility companies globally, amounting to trillions of dollars annually. This pressing issue underscores the need for transformative measures within the electrical grid. Accordingly, our study explores the integration of block chain technology into smart grids to combat electricity theft, improve grid efficiency, and facilitate renewable energy integration. Block chain’s core principles of decentralization, transparency, and immutability align seamlessly with the objectives of modernizing power systems and securing transactions within the electricity grid. However, as smart grids advance, they also become more vulnerable to attacks, particularly from smart meters, compared to traditional mechanical meters. Our research aims to introduce an advanced approach to identifying energy theft while prioritizing user privacy, a critical aspect often neglected in existing methodologies that mandate the disclosure of sensitive user data. To achieve this goal, we introduce three distributed algorithms: lower–upper decomposition (LUD), lower–upper decomposition with partial pivoting (LUDP), and optimized LUD composition (OLUD), tailored specifically for peer-to-peer (P2P) computing in smart grids. These algorithms are meticulously crafted to solve linear systems of equations and calculate users’ “honesty coefficients,” providing a robust mechanism for detecting fraudulent activities. Through extensive simulations, we showcase the efficiency and accuracy of our algorithms in identifying deceitful users while safeguarding data confidentiality. This innovative approach not only bolsters the security of smart grids against energy theft, but also addresses privacy and security concerns inherent in conventional energy-theft detection methods.

## 1. Introduction

Electricity theft presents a severe problem in conventional power systems worldwide, leading utility companies to struggle with substantial financial losses estimated to reach billions of dollars each year [1]. The solution to this widespread problem can be found in the cutting-edge field of smart grids, a revolutionary concept designed to enhance electrical grids’ efficiency, dependability, and sustainability. Smart grids are powered by advanced devices called “smart meters”, which replace traditional analog meters. These devices accurately document users’ energy usage and possess communication functionalities, establishing an interactive connection between utility companies and energy consumers. This bidirectional communication channel offers improved oversight and supervision of power networks worldwide. However, advancement inevitably brings new obstacles. In contrast to mechanical meters, smart meters are vulnerable to a range of assaults, particularly those that exploit network vulnerabilities. This heightened susceptibility exposes the risk of a rise in incidents of energy theft, hence amplifying the seriousness of the issue worldwide. Researchers have investigated innovative methods to detect occurrences of energy theft in traditional power systems [2]. Various methods have tackled the complex task of identifying prospective “energy thieves”, from extreme learning machines to genetic algorithms and support vector machines. Nagi et al. employed a combination of genetic algorithms (GAs) and support vector machines (SVMs) as an alternative to extreme learning machines [3]. The method focuses on optimizing parameters through GAs for improved SVM performance. It enhanced parameter tuning and model robustness, while potential drawbacks involved increased computational complexity. Depuru et al. [4] have also created another strategy that utilizes support vector machines (SVMs) for data mining. Unfortunately, these measures are not reliable when it comes to identifying individuals who steal energy. Bandim et al. propose a strange approach where an integral observer measures the total energy usage of a particular group of users [5]. This unique methodology effectively detects every case of energy theft by comparing the total energy consumption with the reported user data. However, efforts to fight energy theft face obstacles. The utility company’s collection of personal information, such as load profiles and meter readings, raises substantial concerns regarding privacy and safety. The possibility of third parties, such as insurance companies and marketing businesses, using this sensitive data introduces additional complexity to an already intricate situation. To identify the culprits of energy theft, utility companies (UCs) require access to specific consumers’ confidential data, including their load profiles or meter readings at specified times, to successfully carry out the procedures mentioned earlier. The revelation of such information would raise concerns regarding privacy, security, and other related issues, as it would violate users’ privacy. External entities might be interested in acquiring users’ confidential data. Insurance companies can acquire load profiles from the UC to adjust policy rates. They may, for instance, discover patterns of power use that raise the likelihood of fire in a building and then adjust insurance rates appropriately. These data could also be helpful for marketing firms pursuing new clients. Criminals could also use these personal data for illicit purposes. For instance, criminals might figure out how their potential victims usually use energy by studying their consumption patterns. They are so good that they can detect if their intended victim has a burglar alarm [6]. State legislatures and public utility companies are being urged to address the growing concern about consumers’ privacy by numerous researchers like Quinn [7], who have recognized the potential for high-resolution electricity usage data to reveal personal details about consumers’ daily lives and violate their privacy [8]. Modernizing electrical grids has become imperative for addressing various challenges, including electricity theft, grid inefficiencies, and integrating renewable energy sources. In this context, the anti-theft electricity blockchain-enabled smart grid concept has gained prominence [9,10]. This study explores this innovative approach’s underlying principles and implications, drawing insights from various relevant research articles. Blockchain technology has surfaced as a prospective solution to augment smart grids. The fundamental attributes of decentralization, transparency, and immutability of blockchain technology are in accordance with the demands of contemporary grid systems. The study’s authors highlight the potential of blockchain technology in creating a secure and transparent record of electrical transfers within the grid. The ledger, comprised of interconnected nodes in a distributed network, guarantees the verifiability and immutability of every transaction [11,12].

## 2. Related Work

Electricity theft continues to be a major obstacle in conventional grid systems, leading to financial losses for utility companies and severe safety risks. Blockchain-based smart grids provide a strong array of theft prevention methods. Real-time monitoring enables utility providers to identify abnormal consumption patterns rapidly. This feature facilitates the detection of probable instances of theft, prompting fast notifications and inquiries [5,7]. The fundamental transparency of blockchain records poses a significant challenge for malicious entities seeking to tamper with data. Both consumers and utility businesses have access to a common ledger, which fosters transparency and serves as a disincentive against theft. By leveraging blockchain technology, smart contracts may precisely compute invoices by utilizing up-to-date consumption data, decreasing billing disputes and the risk of fraudulent activities. The decentralized nature of blockchain technology allows for peer-to-peer energy trading within the smart grid ecosystem [8,9]. The concept has been examined to understand how blockchain enables direct energy transactions between customers, enabling individuals with surplus energy, such as solar panel owners, to vend their extra electricity to nearby residents. This not only encourages the utilization of sustainable energy sources but also enhances security by transparently and securely recording transactions. Ensuring the security and privacy of data within a smart grid that utilizes blockchain technology is of utmost importance [1,11,12,13]. The integration of blockchain technology into the Internet of Things (IoT) highlights the significance of establishing robust security measures [14,15]. Data transmission and storage are secured using advanced cryptographic algorithms, ensuring the confidentiality and integrity of critical information. Consumer consent procedures are crucial for enabling individuals to regulate access to their data [13,16].

Utilizing blockchain technology in smart grids offers significant advantages in optimizing the grid and enhancing its efficiency. Blockchain technology’s real-time monitoring and predictive maintenance capabilities can mitigate energy losses and minimize downtime. Swift detection and resolution of grid problems and outages can effectively minimize disruptions and enhance the overall reliability of the grid [17,18,19]. Various research papers emphasize practical applications and real-world evaluations of blockchain technology in intelligent power grids. Several authors provide examples of how blockchain has effectively been utilized in the energy industry. The conversation revolves around enhancing transparency and traceability in energy transactions through implementing blockchain technology, mitigating the potential for criminal activity. Incorporating blockchain technology into smart grids presents significant regulatory and economic implications. The need for regulatory frameworks is to adapt to accommodate blockchain-enabled smart grids [14,20,21]. Policymakers need to find a middle ground between promoting innovation and safeguarding the security and privacy of grid operations [22]. Additionally, the economic implications of transitioning to such a system must be carefully evaluated [23,24].

Privacy and security measures are integral to blockchain topology, utilizing public key cryptography to secure transactions. Additionally, techniques like confidential transactions (CTs) encrypt transaction amounts, ensuring privacy by concealing the actual transaction values. These methods, alongside other available blockchain utilities, enhance privacy and safeguard sensitive data in our system.

Insufficient research has been conducted on detecting energy theft in smart grids, focusing on preserving privacy. Similar to prior research on privacy-preserving data aggregation in mobile sensor networks, Li et al. [25] developed a system to collect the aggregate energy consumption of users at a smart grid distribution station. However, electricity exploitation from smart grids is beyond the capabilities of such algorithms. To the best of our knowledge, our examination of the energy theft detection issue considers user privacy. Specifically, understanding a user’s electricity consumption is essential, as has been carried out in earlier works, to determine if the user is engaging in deception; yet this information reveals the user’s privacy [26]. Consequently, it appears that there are two competing concerns: detecting energy theft and protecting users’ privacy. Protecting users’ privacy while detecting energy theft is a difficult task. Here, we provide three distributed algorithms that use P2P computing to find the “honesty coefficients” of the users by solving an LSE [27]. An honest user is one whose honesty coefficient is 1. Otherwise, the user engages in misconduct if their stated energy consumption is less than what they used, as indicated by an honesty coefficient greater than 1. Users’ privacy can be protected since they do not need to provide any information about their energy consumption.

These simulations demonstrate the feasibility of integrating blockchain into the modern grid; hardware implementation faces several challenges. The foremost concern is the reliance on internet connectivity and the need for interoperability between blockchain-compatible smart meters. Adapting traditional smart meters to meet these requirements may necessitate software updates and modifications. Moreover, enhancing hardware capabilities, such as incorporating independent processing units like Raspberry Pi or customized smart meters with sufficient processing power and memory, is essential for efficient blockchain operation. 

The proposed research focuses on detecting electricity theft in blockchain-enabled smart grids, influenced by various important domestic and international trends. On a national level, it corresponds to the continuous modernization of electrical grids, where sophisticated technologies such as smart meters and sensors are being incorporated to improve the efficiency and sustainability of the grid. Moreover, with the increasing use of renewable energy sources in local power networks, the possibility of peer-to-peer energy trading becomes more evident, highlighting the crucial need for smart grids. The research’s main objective is to minimize electricity theft, aligning with governments and utilities’ growing emphasis on energy efficiency and loss reduction. Moreover, implementing blockchain technology in the energy industry, known for its transparency, security, and decentralization, presents an attractive solution for handling transactions in smart grids [13,28]. Despite these challenges, our study focuses on blockchain’s utilization as a cloud-based data compilation and storage center, sidestepping the need for detailed examination of hardware implementation requirements. However, it underscores the necessity of addressing compatibility issues and adapting meters to accommodate blockchain functionality in practical deployments.

Globally, the proposed research aligns with the wider trend of global smart grid activities. Countries are allocating resources to upgrade their electrical infrastructure, frequently incorporating real-time monitoring and advanced metering technologies. The trend of transitioning to decentralized energy systems involves the active involvement of consumers and prosumers in energy generation and delivery [18]. This paradigm is in line with the possibilities of blockchain-enabled smart grids. The issue of energy security is a matter of worldwide importance, and stealing electricity presents financial and security hazards in different parts of the world, attracting international attention. The investigation into the function of blockchain in energy management, namely in terms of improving security and transparency, is apparent through numerous pilot projects and research endeavors. The suggested research is positioned in this particular setting as a leading endeavor that tackles crucial obstacles and prospects in the constantly changing energy sector. As a result, it has become a noteworthy and timely subject of importance, both inside the country and on a global scale [18,29].

This study aims to establish a robust framework that guarantees secure electricity transmission in peer-to-peer transactions inside a decentralized system. The primary objective is to enable completely automated smart contracts between electricity producers, prosumers, and consumers within a smart grid equipped with anti-theft electricity blockchain technology. The crucial components essential for the effectiveness of this method encompass the accurate detection of energy theft and its specific location, strengthening the power grid against unlawful actions.

Moreover, the intended system seeks to provide exceptional dependability and sophisticated infrastructure to improve the overall effectiveness of electricity transactions. An essential element of this framework is its dedication to guaranteeing absolute transparency and security in the transmission of electricity, which promotes confidence among all parties involved. Significantly, the strategy is specifically crafted to be economically efficient, using pre-existing infrastructure and eliminating the need for extra equipment and sensors. Incorporating peer-to-peer energy transactions enhances the system’s autonomy and efficiency, facilitating direct exchanges between producers and consumers. This ultimately strengthens the electrical ecosystem, making it more resilient and secure. Table 1 presents a detailed comparison of the proposed technique and comparison with already carried-out studies.

The remaining part of the paper is structured as follows: the network models are explained in Section 3, while Section 4 delineates the linear system of equations utilized to detect energy theft. The comprehensive system modelling is elaborated upon in Section 5. Section 6 details the proposed distributed algorithms for solving the LSE. The results of the simulation are detailed in Section 7. A summary of the paper’s findings is provided in Section 8.

## 3. Modelling of Networks

This section commences with an introduction to the network architecture discussed in this paper. Following that, a brief summary of potential energy-theft attacks on smart meters (SMs) and potential implementations of the proposed algorithms in smart meters is presented.

### 3.1. System Architecture

Communications and electricity networks are superimposed in the smart grid. Utility companies use control centers to supervise their distribution stations and distribution networks. They also install smart meters at users’ residences to quantify their power consumption. To ensure effective communication between users and customer centers, especially considering their frequent geographical distance, a crucial component called “the collector” is deployed in each service area to facilitate the flow of information. At each collector, one SM is installed to determine the overall energy consumption of the serviced area. The network architecture that is commonly observed is illustrated in Figure 1. The collector and the users’ SMs comprise a neighborhood area network (NAN) in a serviced area. The SMs utilize wireless communication for interactions among themselves and with the collector. In contrast, communications between the CC, the DS, and the collector are facilitated through wired connections. 

### 3.2. Security Incidents Targeting Smart Meters

Smart meters are capable of offering consumers an abundance of unique benefits. Users may, for instance, be informed of the real-time cost of electricity, enabling them to schedule the operation of specific electrical appliances accordingly. Incentive-based load reduction signals can also be transmitted by smart meters to users, compensating them for their energy conservation efforts. In contrast to mechanical meters, which are solely susceptible to physical tampering, smart meters are susceptible to a wider variety of attacks. This increased vulnerability may facilitate energy theft, rendering it an even more significant concern within smart grids.

SMs and conventional mechanical meters are at risk of this attack. It pertains to situations in which unauthorized individuals physically tamper with their electricity meters to record inaccurate readings to reduce their electric bills. Attacks on electricity meters encompass various actions such as meter reversal, strong magnet disruption, pressure coil damage, supply-voltage regulation manipulation, and meter disconnections. An approach to identifying attacks physically involves conducting a visual inspection of the meter for indications of damage, such as ruptured seals [37]. However, this detection method is time-consuming and resource-intensive because UC personnel must physically inspect the meters on the user’s premises. Additionally, damage may not be readily apparent, and seals might require replacement.

An unauthorized user can launch a network attack by operating a malicious node. Unauthorized users may, for instance, impersonate their own SM to cause it to exhibit abnormally low energy consumption. Network-based attacks may be more challenging to detect and simpler to initiate. Aside from challenging the accuracy of their smart meter readings, users may obtain unmeasured energy via a conductor that circumvents the meter. In this instance, the smart meter inaccurately registers the user’s energy consumption and, as such, may also be deemed to be under threat. The algorithms that are proposed can resolve each of these issues.

The algorithms that have been proposed are receptive to firmware implementation in smart meters. Numerous safeguards for embedded system firmware have been suggested, including password protection, centralized and local intrusion detection, self-reporting of intrusions, and centralized intrusion detection. An example is the remote attestation mechanism developed by LeMay et al. [38], which enables centralized intrusion detection of every SM within a given neighborhood. The UC will identify any firmware intrusion and deem the suspect SMs untrustworthy; therefore, they must be inspected. As a result, it can be relied upon that SMs will accurately implement the suggested algorithms for detecting energy theft which will protect privacy.

## 4. Linear Equation System for Detecting Energy Theft

This section introduces a mathematical model designed to identify energy theft. As previously stated, it is assumed that an SM is deployed at the collector to provide the collector with information regarding the aggregate energy usage of the consumers within the service area. Additionally, it is presumed that the SM is installed at each user’s location by the UC and can capture energy consumption in real time.

Consider a set of n users experiencing a NAN scenario. The time interval for sampling is denoted by the symbol SP. After each sampling period, all n + 1 SMs will record their energy consumption from the preceding sampling period. The energy consumption measured by the collector at time ui and user j (1 ≤ j ≤ n) is denoted by pui, j and Pui, respectively. A further definition of an honesty coefficient is provided for each user j, represented by kj where kj > 0. Therefore, the actual energy consumption of user j from time instant ti → SP to time instant ti is denoted by kj · p_ui,j_. Considering that the combined actual energy usage of all individuals in the preceding sampling interval should align with the neighborhood’s overall energy consumption recorded at the collector at time ti, it can be inferred that:(1)$$k_{1Pu,1}+k_{1Pu,2}+\ldots k_{nPu,n}={\tilde{P}}_{ti}$$

Our goal is to identify the values of kj’s.

If

kj = 1, it indicates that user j is honest and not engaged in energy theft;kj > 1, it indicates that user j is not honest and is stealing energy;0 < kj < 1, it indicates that user j has recorded more consumption than they actually consume, suggesting a potential malfunction in their smart meter.

Specifically, when n linear equations are involved, the resulting system of equations (LSE) can be expressed as follows:(2)$$k_{1Pt1,1}+k_{1Pt1,2}+\ldots k_{nPt1,n}={\tilde{P}}_{ti}\;k_{1Ptn,1}+k_{1Ptn,2}+\ldots k_{nPtn,n}={\tilde{P}}_{tn}$$

Expressing the concept in matrix notation is another way to articulate it:(3)$$Pk=\tilde{P}$$

The information captured and saved by user j, or SMj, is indicated in the jth column of P. Meanwhile, the data documented by all users at time ti is depicted in the ith row of P. The collector is permitted to select n time instances in which each Pti has a unique value. In cases where the value of n is significant, there is a high likelihood that independence exists between the least squares estimate and the rows of P. Consequently, when this occurs, the feasible solution kj = p_ti,j_/p_ti,j_ prevails as the sole unique solution to the described LSE. Here, p_ti,j_ denotes the actual energy consumption of user j during the time interval from ti–SP to ti.

It is important to acknowledge that our model fails to incorporate energy dissipation, also known as technical losses, within the electrical power system. These TLs primarily arise from the inherent defects present in low-voltage power lines and transformers. Nevertheless, TLs can be computed independently of energy measurements taken by consumers. As an illustration, Oliveira et al. [39] establish a method for computing TLs by leveraging insights obtained from the distribution network and measurements taken at the distribution station, a process that ensures the protection of user privacy. Consequently, once the collector calculates the technical losses, it becomes feasible to adjust the model by subtracting the computed TLs from vector $\tilde{P}$.

Furthermore, it is important to highlight the fact that the calculation of the honesty coefficient vector, denoted as k, remains unaffected by delays. Alternatively stated, the discovery and real-time transmission of k to the collector are not prerequisites. This emphasizes the importance of electricity pricing, incentive-driven load-reduction signals, emergency load-reduction signals, and other real-time communication within the NAN.

## 5. Materials and Methods

The research proposes integrating blockchain technology, specifically Blockchain v2.0, into the smart grid infrastructure. Blockchain’s inherent features, such as decentralization, transparency, and immutability, make it a suitable candidate for enhancing security and reliability within the grid [11,12]. A significant innovation proposed by this research is using smart contracts within a blockchain-enabled smart grid. These smart contracts serve as agreements and transaction facilitators between consumers, prosumers, and pure producers. The smart contracts are designed to be fully automated, transparent, and secure, allowing for peer-to-peer transactions without the need for third-party intermediaries [13,40,41,42]. The block diagram of the methodology is shown in Figure 2:

Figure 3 shows a smart grid’s distribution network model of consumers, producers, or pure producers (such as national grid and private producers). The below-mentioned model has the advanced metering infrastructure (AMI) of a smart grid network. Although the model is not decentralized, it means a third party is involved in peer-to-peer trading and is not fully automated with slow means of transactions. To identify energy theft in a typical smart grid network, more equipment and sensors would be needed at every node and pole, and advanced algorithms would be needed.

Figure 4 shows the blockchain-enabled smart grid. The model shows that each electricity user has a smart contract within peer-to-peer energy trading. These smart contracts will be fully automated. The theft detection in this network system will not require much equipment and many sensors, compared to other proposed or researched theft-detection methods for smart grid networks. This proposed research will identify theft at the distribution lines through smart contracts, which will make energy transactions secure for both parties in P2P energy trading.

Figure 5 shows energy trading through a smart contract. The above figure shows the energy trading between two parties in a blockchain-enabled smart grid with the role of a smart contract. The consumer is giving demand to the producer, and the producer is transmitting the required power to the consumer; not only this, but both users are confirming the transactions of energy at every second. This means that only the required energy will be uploaded to the network, and will make a successful transaction. Suppose the consumer does not receive the demanded power or unbalanced power in the distribution network. In that case, the smart contract will identify the energy theft in the network, and the smart contract will automatically inform the network service provider. The network service provider will take the required actions against the theft.

The location of the theft at any phase of a transformer (node) will be identified by installing a single three-phase load sensor at the output of the node (transformer). As the load data of all the connected users are already recorded with the help of smart contracts, any theft (either soft- or hard-line theft) occurring will be identified at every second. A state channel is a two-way communication channel that allows us to make a group of transactions back and forth; we will then obtain the eventual result and then put it on the main blockchain.

Figure 6 shows that power metering has been carried out at both peers. The consumer will show the demand per second/live, and the prosumer or producer will allow the demanded power to be given to the consumer. Communication in such peer-to-peer transactions will be carried out on the side chain as “layer 2” of Ethereum 2.0, until the completion of a smart contract. After completion of a successful transaction, the result will be loaded to “layer 1” as the main blockchain. We will consider scenarios involving various numbers of users (15, 30, 50 and 100) and different probabilities of energy theft (0.3 probability). The simulated data will include honesty coefficients for illegal users and other relevant parameters.

An assessment of various algorithms, including those based on (but not limited to) LU decomposition (LUD), LU decomposition with partial pivoting (LUDP), and optimized LUD composition (OLUD) for large clusters, will be conducted across different scenarios. These scenarios will involve variations in the number of users and probabilities of energy theft. The study conducts simulations using the specified scenarios and algorithms. The simulations are based on the generated data. The simulation will be prepared in MATLAB. Table 2 compares these existing and newly proposed optimized LUD algorithms.

## 6. Determining Honesty Coefficients through P2P Computing

We present three algorithms that employ P2P computing, also referred to as distributed or collaborative computing, to address the linear system of equations in (4)–(6) while ensuring user privacy. The challenge arises from the necessity for each smart meter SMj to ascertain its honesty coefficient kj without accessing the recorded energy consumption data p_ti,l_ from other smart meters. Here, the parameters satisfy 1 ≤ i ≤ n and j ≤ l.

We first introduce an approach called LUD, leveraging LU decomposition, to detect energy violators while ensuring user privacy. It has come to our attention that the original form of LUD may lack numerical stability when applied to large networks. When n is substantial, the inaccuracies inherent in the use of finite resolution numbers may result in solutions that contain substantial errors. Consequently, subsequent to the introduction of the LUD algorithm, we devise an additional algorithm known as improved LUD with partial pivoting (LUDP), which accomplishes numerical stability through the exchange of rows of the matrix P during LU decomposition. Following that, we propose OLUD, an algorithm based on LUD, which exactly follows the procedure of LUD but the partial decomposition part is excluded, due to the fact that it enables stable peer-2-peer computing in large-scale networks in order to identify energy criminals. We outline each of these three techniques in turn, first for situations where the honesty coefficient vector, k, remains constant, and then for situations where it evolves over time.

### 6.1. The LUD Algorithm

Let us commence by elucidating the LUD algorithm, which is built upon distributed LU decomposition. The LU decomposition entails the partitioning of the energy consumption data matrix P into two triangular matrices, designated as P *=* LU. This decomposition comprises a lower triangular matrix *L* and an upper triangular matrix U. The computation of the elements within the upper triangular matrix U is outlined as follows:(4)$$u_{i^{\prime }j}=0,i>j$$
(5)$$u_{1,j}=p_{t1,j},,j=1,2,\ldots ,n$$
(6)$$u_{r,j}=p_{tr,j}-\sum_{k=1}^{r-1}l_{r,}kUk,j,r=2,\ldots ,n,j=r,\ldots ,n$$

In matrix P, the element p_ti,j_ represents the i_th_ element of column j. Additionally, the elements of the lower triangular matrix L can be obtained through the following derivation:(7)$$L_{i,j}=0,i<j$$
(8)$$l_{i,1}=\frac{p_{ti,j}}{p_{t1,j}},i=1,2,3,\ldots n-1,,n$$
(9)$$l_{i,q}=\frac{p_{tr,q}-\sum_{k=1}^{q-1}l_{i,}kUk,q}{u_{q,q}},\;\;\;\;\;\;q=2,\ldots ,n,i=q,\ldots ,n$$

It is important to highlight that the diagonal elements of matrix L are all set to 1. This condition guarantees the uniqueness of the decomposition of matrix P. Once matrices L and U have been calculated together, the subsequent system can be solved:(10)$$yL=\tilde{P}$$
(11)$$Uk=\tilde{y}$$

To determine the value of y, each preceding SM_j−1_ will compute $y_{j}$ using the expression $y_{j}$ = ${\tilde{p}}_{tj}$,
(12)$$y_{j}={\tilde{p}}_{tj}-\sum_{q=1}^{j-1}j,qYq,\;\;\;\;\;\;j=2,\ldots ,n$$

The values needed for this calculation are the elements of y in row j of L with an index less than j. Lastly, make sure that every SM y in row j of L has an index lower than j. As a last step, each SM_j_ finds k_j_ by applying backward substitution, so that $k_{n}=\frac{y_{n}}{u_{n,n}}$.
(13)$$k_{j}=\frac{y_{j}-\sum_{p=j+1}^{n}u_{j,}pkp}{u_{j,j}},\;\;\;\;\;\;j=1,\ldots ,n-1$$

Hence, our LUD algorithm is split into two sections, Procedure 2 detailing backward substitution and Procedure 1, detailing distributed LU decomposition. Also, the collector needs to assign a number between 1 and n to each SM before the algorithm can start, and no one other than the SM and the collector knows its index number.

The collector also sends P_tj+1_ to each SM_j_ so that the SMs can work together to calculate L, U, and y. At the collector, there is a smart meter that we will call SM_0_. In response to a control message from the collector, all SMs initiate Procedure 1.


**Procedure 1: Distributed LU Decomposition and how it works:**



**Input:**


*j*: the current processor number.Ptj + 1: a partial pivot matrix that is sent from processor j to processor j + 1 in step 14.Steps:If j = 0 or processor j receives packets from processor j − 1, then oIf j = 0, then processor 1 computes y₁ using Equation (12) and transmits it only to processor 1.oIf 1 ≤ j ≤ n − 1, then ▪For q = 1 to j, processor j computes uq,j using Equation (3).▪For q = j + 1 to n, processor j computes lq,j using Equation (6).▪Processor j computes yj + 1 using Equation (9).▪Processor j sends columns 1 through j of L, along with all previously known elements of y1 through yj + 1, exclusively to processor j + 1.oIf j equals n, processor n informs the collector about the availability of L, U, and y.


**Algorithm description:**


The algorithm works by distributing the matrix A among the processors. Each processor is responsible for computing a portion of the L and U matrices. The processors engage in communication with one another to facilitate the exchange of data throughout the computation process. The algorithm is divided into three main phases:Phase 1: In this phase, the processors compute the first column of L and the first row of U.Phase 2: In this phase, the processors compute the remaining columns of L and the remaining rows of U.Phase 3: In this phase, the processors compute the y vector.

The algorithm uses a pipelined approach, in which each processor works on a different part of the computation at any given time. Enhancing the algorithm’s efficiency is achieved by minimizing the necessary communication. The flow chart of recording data for electricity transaction is illustrated in Figure 7.


**Procedure 2: Backward Substitution**


This algorithm is employed for solving a system of linear equations in which the coefficient matrix is in upper triangular form. The procedure involves the following steps:If j = n or the source machine (SM) receives a packet from SMj + 1, then proceed to step 2.Calculate kj according to the formula in Equation (13), utilizing SMj + 1 if required.Determine E(kj) and send the computed value of E(kj) to the collector.If j is not equal to 1, then proceed to next step.Compute uqjkj for q = j − 1, j − 2,−1,…0,1.Compute sj = j uj − 1, qkq and transmit sj to SMj − 1.Transmit U1.jkj, U2.jkj, …, Uj − 2.jkj, U1.j + 1kj + 1, U2.j + 1kj + 1, …,Uj − 2,j + 1 kj + 1, …, Unkn, …,Uj − 2.nkn to SMj − 1.End if.End if.

The backward substitution algorithm is an efficient way to solve upper triangular systems of equations. It surpasses Gaussian elimination in efficiency, offering a more effective approach for solving systems of linear equations. The backward-substitution algorithm is also more stable than Gaussian elimination, meaning that it is less likely to produce inaccurate results due to rounding errors. The backward-substitution algorithm can be used in various applications, such as solving systems of equations that arise in circuit analysis, control theory, and numerical analysis. It is also used in some cryptography algorithms.

### 6.2. The LUDP Algorithm

LUD may lack numerical stability for significant values of n, as mentioned earlier. This paper introduces an alternative method known as LUDP designed to address stability concerns. The algorithm involves reordering rows in matrix P during partial pivoting, ensuring that the element with the highest absolute value in each column is strategically placed in the diagonal position of the matrix. Consequently, the LUDP decomposition can be expressed as EP = LU, where the permutation matrix is denoted by E.

The LUDP algorithm includes forward substitution, backward substitution, and LU decomposition with partial pivoting. LU decomposition with partial pivoting is shown in Procedure 3. We start with U = P, SM1, then update the first column of U by finding the largest element in column 1, setting the row in which it appears as the pivot index, and swapping it for the element in row 1 if it is not there. SM1 also calculates L’s starting column and sends it to SM2 with column 1’s pivot index. LUDP computes U and L differently than LUD, as shown in Procedure 3. This adjustment allows partial pivoting [27]. Upon receiving data, SM2 reproduces SM1’s row interchange by swapping column 2, row 1 of matrix U, with the row containing the pivot index for column 2. Subsequently, SM2 updates the second column of U, conducts its own row interchange by exchanging the maximum element in column 2 of U with the second element, and establishes the pivot index of column 2 as the row where the maximum element is positioned. After computing the second column of L, SM2 transmits to SM3 the initial two columns along with the pivot indices from both SM1 and SM2. After receiving columns 1 to n − 1 of L and pivot indices from SMn − 1, SMn determines column n of U. All previous row interchanges are repeated and its own row interchange is performed. Consider l_(j,j) = 1 when 1 ≤ j ≤ n.

Due to Equation (4), P’s row interchanges must match P’s. Thus, we let SMn send the indices of all n pivot rows to SM0, which exchanges P rows identically. We can now calculate y and k using Equations (6) and (9). Because y is determined according to Equation (12), Procedure 3 must finish before calculating it. In contrast, LUD calculates y simultaneously with L and U. We recommend the forward substitution procedure in Step 4, allowing smart meters to collaboratively solve for the variable “y”. Unlike backward substitution, forward substitution starts at SM1 and computes yj according to Equation (12). Finally, back substitution, as discussed in Procedure 2, can solve for k with y.


**Procedure 3: LU Decomposition with Partial Pivoting (LUDP)**


Input: The procedure takes an input of j, which represents the current column of the matrix U being processed.Initialization:oSet U = P, where P is the permutation matrix obtained from the previous column (j − 1).oIf this is the first column (j = 1) or no pivots have been received from the previous column.Pivot Row Selection:oIf not the first column (j ≠ 1), iterate through all previous columns (f = 1 to j − 1). Receive the l elements (column entries) and pivot row indices from the previous column (SM(f − 1)).Modify the present column (j) of matrix U by considering the possibility of swapping the jth element in row f with the jth element in the pivot row of SM(f − 1).Update the remaining elements of row r (r = f + 1 to n) in U by subtracting the product of the corresponding l element from the previous column and the jth element of the pivot row in SM(f − 1).oSet l(j,j) = 1 (diagonal element of L is always 1).oIf not the last column (j ≠ n), determine the pivot rows for the current column (j) of U: Find the element in U with the largest magnitude in the current column (j).Swap the jth element of the row containing the maximum element with the jth element of the current row.Update the l elements in the current column (j) of L corresponding to the new pivot row.Set all other elements in the current column (j) of U to zero.oTransmit the l elements (column entries) and pivot row indices for the current column (j) to the next processing unit (SM(j + 1)).Finalization:oIf this marks the final column (j = n), inform the collector that the L and U matrices are ready for use. oForward all indices of the pivot rows to the collector.


**Procedure 4: Forward Substitution**


Procedure 4 iteratively computes values in a vector y using a specific algorithm, likely involving a matrix L and a vector Pt. These computations are distributed across multiple processing units called Sensor Modules (SMs).

Algorithm:Initialization:oIf this is the first SM (j = 0): Compute y1 = Pt1.Transmit y1 to the next SM (SM1).Iteration:oFor intermediate SMs (1 ≤ j ≤ n – 1): Compute yj + 1 using Formula (13) and potentially sj − 1 (not provided in the image).Transmit yj + 1 to the next SM (SMj + 1).Calculate lq,jyj for q = j + 1, j + 2, j + 3 …, n – 1, n.Calculate sj = sum(lj + 1,qyq) for q = 1 to j.Transmit sj and lj + 2,1y1, …, ln,jyj to the next SM (SMj + 1).Finalization:oIf this is the last SM (j = n): Notify the collector that y is fully computed.

Similar to LUD, adjacent smart meters (SMj and SMj + 1) can produce symmetric security keys and encrypt data transmitted in Procedure 2, protecting user privacy. The LUDP algorithm takes longer than LUD. In LUDP, forward substitution for y computation is only possible after L and U are achieved. In contrast, in the LU decomposition, the values of y, L, and U can be determined simultaneously. Unlike LUD, LUDP is characterized by numerical stability.

### 6.3. The Optimized LUD Algorithm

Figure 8 explains the working flowchart of our developed OLUD algorithm, which is an advanced, optimized algorithm for detecting energy theft and protecting privacy. The implementation of OLUD on block chain infrastructure is a secure, transparent, and open form of data collection and evaluation. All the users (smart meter owners) can identify their own data in the block chain as a client. They are only able to identify themselves, and no one else, because of their private key. All block chains are a form of cloud computing, but their coding and structural algorithms make them unique and secure. The bigger the group of SM (smart-meter) users in a block chain cluster, the more secure it is, but the drawback is that it will take longer to update and modify the cluster. The computation complexity of this algorithm has been displayed in the formula. The block chain users calculate their honesty coefficient and the coefficient of every other user; this is repeated for all the users in the block chain. Thus, a big block chain for thousands of users will start to become progressively slower. The users will keep each other honest; thus, even if a user were to manipulate their SM data, the other users would detect abnormal behaviors and trigger a tampering alarm caused by a non-matching honesty coefficient. The weaknesses of the simple LUD and its various variants become more apparent with larger datasets, resulting in slower performance and an increased likelihood of error. We have overcome this by reducing complexity and avoiding extra steps, such as partial pivoting or full pivoting. By limiting the honesty coefficient to a maximum value of 2, we restrict the need for excess computation and allow for accurate theft detection for small and large groups of users.

LUDP needs partial pivoting with forward substitution and backward substitution. This process is reiterative and needs a lot of computation power as the number of users increases. But if we restrict the maximum variation in LUD of the honest coefficient to a maximum value of 2 between content and variable theft patterns, we can introduce a stable LUD method with complexity comparable to the basic method but with a stable working area of LUDP without its complexity.

### 6.4. Variable Honesty Coefficients

The algorithms mentioned earlier, namely LUD, LUDP, and OLUD, have exclusively presumed that the honesty coefficient vector k remains constant. Nonetheless, the rate at which an unauthorized user inappropriately acquires energy may exhibit variability. Put differently, an unauthorized individual has the ability to modify the smart meter to extract energy at varying rates and during varying periods. If k is altered in an LSE, the proposed algorithms may become inoperable. Following this, adaptive algorithms are developed to tackle this issue. It is observed that the previously mentioned algorithms produce an honesty coefficient vector wherein a significant number of elements do not equal 1, whenever k is altered in an LSE. Therefore, upon obtaining the vector k representing the honesty coefficient and tallying the elements whose values do not equal one, the collector can deduce, through statistical analysis, whether such a large number of energy robbers are feasible within the network. If the occurrence of this event is deemed improbable, the collector has the option to reduce the sampling period (SP) and repeat the algorithms until the probability of the event becomes significant, while maintaining a constant value for k. The mathematical model is presented as follows. Consider a serviced area with n users, all of whom independently commit energy theft with an equal probability denoted as p. The variable X represents the total number of individuals engaging in energy theft within the neighborhood, and it follows a binomial distribution at that specific point.

By executing LUD/LUDP/OLUD, the collector acquires “k”, which enables it to determine the quantity of elements whose values do not equal one; this quantity is represented by Y. The collector can then calculate the likelihood of this event happening using the following method:(14)$$P\left(X=Y\right)=(\begin{array}{c} N\\ y \end{array})P^{y}{\left(1-P\right)}^{N-y}$$

Furthermore, X remains a random variable if each user j independently commits energy theft with a unique probability p_j_; however, its expectation is transformed into E(X) = n_j=1_ p_j_. Consider the Chertoff bounds from [43]:

For ∂>0,
(15)$$P(X>\left(1+\partial \right)E\left(X\right))<e^{-E(X)f(\partial )}$$
where $f\left(\partial \right)=\left(1+\partial \right)\log\left(1+\partial \right)-\partial$.

For 0<∂>1,
(16)$$P(X<\left(1-\partial \right)E\left(X\right))<e^{-\frac{1}{2}E(X)\partial^{2}}$$

Subsequently, the collector may deduce the accuracy of the acquired k through calculation.
(17)$$P\left(X\geq Y\right)<e^{-E(X)f(\partial )}with\partial =\frac{y}{E(X)}-1$$

When Y > E(X), and
(18)$$P\left(X\leq Y\right)<e^{-\frac{1}{2}E(X)\partial^{2}}with\partial =1-\frac{y}{E(X)}$$

In cases where the value of Y is less than the expected value of X (E(X)), and in situations where Y is equal to E(X), the probability of X being equal to Y is established as 1^6^.

Accordingly, should the calculated probability P fall below a specified threshold, the sampling period SP undergoes a reduction by a positive step variable g. Subsequently, the LUD/LUDP/OLUD algorithm is re-employed to obtain an extra set of *k*. This iterative procedure continues until P exceeds the threshold and the resultant k remains unchanged from the previous iteration. At this point, we consider k, representing the actual integrity coefficient vector in the network, to be accurate. Procedure 5 sums up the adaptive LUD/LUDP/OLUD method, which uses varying honesty coefficients to identify unauthorized users.


**Procedure 5: Adaptive LUD/LUDP/OLUD Algorithm**


This approach enhances the effectiveness of the LUD/LUDP/OLUD algorithms within the framework of secure multi-party computation (SMC). These algorithms are utilized to break down a matrix into triangle elements, which is a prevalent process in numerous SMC jobs. The adaptive LUD/LUDP/OLUD algorithm functions by dynamically modifying the sample period in response to the quantity of unauthorized users present in the system. The objective is to minimize the sample size required while maintaining algorithmic accuracy.

Here is a breakdown of the procedure:Repeat:This is the main loop of the algorithm. It will continue to iterate until the stopping conditions are met.The collector directs all SMSs to acquire a specified number of samples using an initial sampling period SP.During each iteration, the collector directs all secure multi-party computation servers (SMSs) to obtain n samples. The frequency of sampling is determined by the initial SP.If the collector acquires every element within set K, then:This conditional statement checks if the collector has received all the elements in the vector k. Vector k contains information about the number of illegal users detected by each SMS.Run the LUD/LUDP/OLUD algorithm:If all elements in k are received, the collector runs the chosen matrix decomposition algorithm (LUD, LUDP, or OLUD) on the collected samples.Y represents the count of elements in set K that are not equal to 1, specifically denoting the number of unauthorized SMSs.The collector calculates the number of illegal SMSs by counting the number of elements in k that are not equal to 1. A value of 1 indicates a normal SMS, while any other value indicates an illegal SMS.end if:This ends the conditional statement.The collector assesses the likelihood of the existence of Y unauthorized users, represented by P.The collector uses the above equations from the original paper to calculate the P that there are Y illegal users. These equations are based on statistical analysis of the sampling data.if P ≥ Pth (a threshold) or P = 1 then:This conditional statement checks if the calculated probability P is greater than or equal to a pre-defined threshold (Pth), or if P is equal to 1.SP–g (g > 0 is a step variable):If the condition in step 8 is met, the collector reduces the sampling period SP by a step value g. This means that the SMS will take samples more frequently in the next iteration.end if:This ends the conditional statement.until Pth and k does not change any more:The loop continues until two conditions are met:oThe calculated probability P < PTH.oThe vector k no longer changes from one iteration to the next. This indicates that the system has stabilized and the number of illegal users is no longer changing.

Output: the output of the procedure is the decomposition of the matrix, as well as an estimate of the number of illegal users in the system.

## 7. Results and Discussion 

Specifically, we conduct two sets of simulations in order to assess the effectiveness of our LUD, LUDP, and OLUD algorithms for detecting energy theft while maintaining the confidentiality of the information. In the first section of this discussion, we discuss the ideal and real-world scenario k factor of theft detection, and as a result, their honesty coefficients are continuous. In the second part of this discussion, we take into account the fact that unauthorized users have their honesty coefficients vary. In addition, we compile data on the energy usage of users, using a collection of data from [43]. Assessments are carried out in which both residential and commercial users are measured at regular intervals. Both of these parts utilize these metrics to illustrate common daily user load patterns across different days of the week and at various times throughout this year. 

### 7.1. Constant Honesty Coeffcient

To begin, we simulate situations in which unauthorized users dispose of energy at a steadily increasing rate. To put it another way, every unauthorized SM chooses a rate at which to steal energy, and they never change this rate or stop stealing energy; hence, the honesty coefficient does not change. To evaluate the effectiveness of LUD, LUDP, and OLUD, the following conditions are taken into consideration: the chance of energy theft committed by each user is 0.3, and the total number of users is 10 and 20 users in the first scenario, which is simulated through the LUD algorithm.

The selection of an honesty coefficient is carried out in a manner that is both random and consistent across all energy thieves [1.1, 10]. Table 3 contain the elements of k for ideal-system versus real-world scenario. As can be seen in Figure 9, the LUD algorithm demonstrates good performance when there are a total of twenty users. According to the data presented in 1 and 3, there are eight users who have an honesty coefficient that is greater than 1. The fact that this is the case suggests that the six SMs document a tiny amount of the energy that they spend. By making use of these discoveries, the collector is able to easily identify those individuals who are wasting energy and determine the precise amount of decrease that they have experienced in their monthly power bills. A further point to consider is that it is obvious that legal consumers have an honesty coefficient of 1, which indicates that they are easy to identify. The results of a sample size of twenty users are depicted in Figure 9, which represents comparable outcomes. The orange line represents ideal k means and blue lines show the actual k means. From this figure we come to know that the users whose actual k is exceeding +1 are actually committing energy theft; on the other hand, users having a k value less than 1 are selling energy back to the grid. In addition, it is possible to observe that LUDP produces results that are comparable to those of LUD in the cases that were discussed earlier. Additionally, the results of OLUD are depicted in Figures 14–17, and include user counts of 20, 30, 60 and 120 users, respectively.

Whenever this takes place, the LUD algorithm displays signs of instability. There are eight illegal users that have been recognized. However, the OLUD algorithm is able to identify each of the individuals who are legally permitted to access the system, precisely.

### 7.2. Comparison between LUD, LUDP, and OLUD

As a result of our analysis of the results using the LUD algorithm, we have discovered that the LUD algorithm is able to detect theft users perfectly when there are ten users. However, when the number of users increases, such as when there are twenty users, the algorithm becomes unstable, fails, and generates some random results that do not exist in real-world work scenarios.

Figure 10 and Figure 11 demonstrate that out of a group of ten users, there are two individuals who have been identified as being dishonest. For the same scenario, when it is interpreted by LUD it is unstable, but now, when LUDP is applied to that system, it works flawlessly and detects six users who are dishonest. Therefore, LUDP is an excellent choice for a network that contains a large number of system users Figure 12 and Figure 13. However, the problem with this algorithm is that it is too complicated. If we talk about our proposed algorithm, which is called OLUD, we will discover that this updated algorithm has the ability to handle vast networks, and it does not need complicated calculations. Figure 14, Figure 15, Figure 16 and Figure 17 make it abundantly evident that OLUD is capable of detecting all theft users in an impeccable manner; nevertheless, it does have a drawback in that it only detects the corrupt user, and the quantity of the k factor is not exactly defined in adequate detail. Because of this, we are able to detect theft in big, complex networks with the assistance of our streamlined algorithm, which requires fewer complicated computations.

In Figure 10, Figure 11, Figure 12, Figure 13, Figure 14, Figure 15, Figure 16 and Figure 17, 
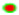
 represent detected users, and 
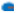
 represent normal users.

### 7.3. Variable Honesty Coefficient

Subsequently, simulations are conducted to examine instances where individuals without authorization illegally acquire energy at different rates. The hypothesis states that once a predetermined amount of time has passed, every energy thief will choose a new honesty coefficient in a manner that is both uniform and random, and it will fall somewhere in the range of [1.1, 10]. First, we assume that the chance of energy theft is the same for all users (*p* = 0.3). Then, we assume that each user’s energy-theft probability is independently and arbitrarily selected from 0.3 to 0.7.

To be more specific, the adaptive LUD method becomes unstable when the number of users in the network exceeds 15. This occurs when the likelihood of deception is equal for all users (*p* = 0.3). The outcomes that were a direct result of the limitations imposed by space are not included. As an additional point of interest, we present the results of the OLUD algorithm for 20, 30, 60, and 120 users, respectively, in Figure 14, Figure 15, Figure 16 and Figure 17. It should be no surprise that every energy thief has been found. Every user engages in energy theft independently and randomly, with a probability ranging between 0.3 and 0.7. In such situations, it has been discovered that the OLUD algorithm can detect fraudulent users effectively and efficiently.

The proposed system addresses scalability challenges in large-scale smart grid deployments by leveraging proven block chain-optimization techniques. In future, to minimize latency, different methods will be employed in block chain systems with millions of users utilized, focusing on factors such as block time, network congestion, and consensus mechanisms. Off-chain transactions and layer-2 scaling solutions, including state channels, sidechains, and rollups, are implemented to reduce latency by executing transactions off the main block chain and aggregating multiple transactions. Additionally, optimizing the consensus mechanism, such as transitioning from proof of work (POW) to faster alternatives like proof of stake (PoS) or practical Byzantine fault tolerance (PBFT), further mitigates latency issues. These strategies ensure efficient transaction throughput and scalability, addressing the demands of large-scale smart grid deployments.

## 8. Conclusions

The urgency for electrical grid upgrades is paramount, especially to address critical issues like electricity theft, grid inefficiencies, and the integration of renewable energy sources. Our comprehensive study explores the potential of block chain-enabled smart grids as an innovative solution to these challenges. We focus on the core principles and implications of block chain technology, emphasizing its suitability for modern grid systems through its decentralization, transparency, and immutability attributes. The deployment of block chain technology in smart grids offers a secure and transparent ledger for grid transactions, enhancing the integrity and verification of each transaction. Our research introduces three pioneering algorithms—LUD, LUDP, and OLUD—tailored toward peer-to-peer computing in these advanced grids. Each algorithm, based on LU decomposition, has its unique approach, with OLUD employing a forward technique and LUD employing a backward one, resulting in OLUD’s superior performance in detecting a larger number of fraudulent users. We prioritize user privacy in our approach, ensuring the confidentiality of personal data and addressing significant privacy concerns that often arise in conventional energy-theft detection methods. Our analysis extends to these algorithms’ computational and communication complexities, noting that OLUD exhibits higher complexity than its counterparts. The adaptive LUD method becomes unstable when the number of network users exceeds 15, particularly when the likelihood of deception is uniform across users (*p* = 0.3). Results of the OLUD algorithm for 20, 30, 60, and 120 users demonstrate its effectiveness in detecting energy theft, as every fraudulent user is identified, with probabilities ranging between 0.3 and 0.7. Through detailed simulations, we demonstrate that these algorithms can effectively identify fraudulent activities, whether constant or variable, by analyzing users’ honesty coefficients. This research not only contributes significantly to the advancement of smart grid technology but also opens new avenues for future exploration in grid security and the protection of user privacy. 

## Figures and Tables

**Figure 1 sensors-24-01668-f001:**
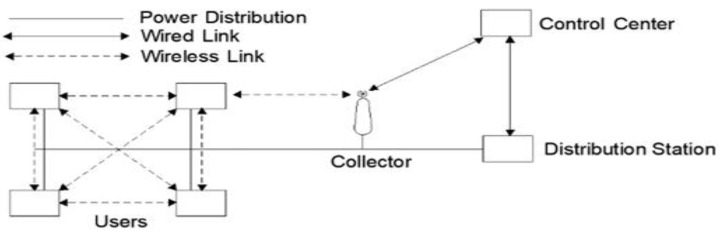
The standard structure of a Neighborhood Area Network (NAN).

**Figure 2 sensors-24-01668-f002:**
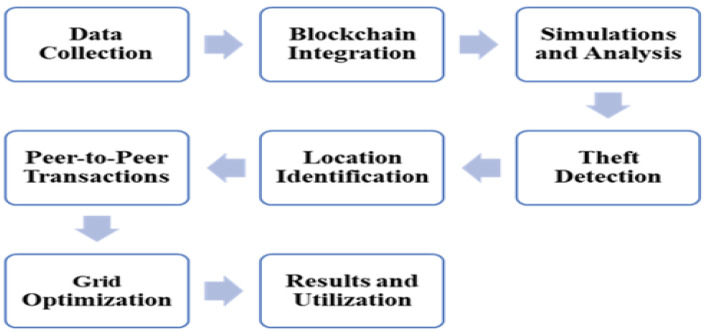
Methodology framework.

**Figure 3 sensors-24-01668-f003:**
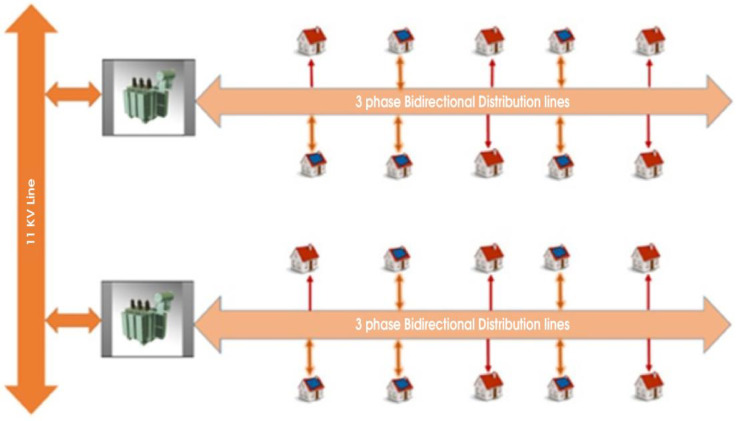
Street model of smart/micro grid.

**Figure 4 sensors-24-01668-f004:**
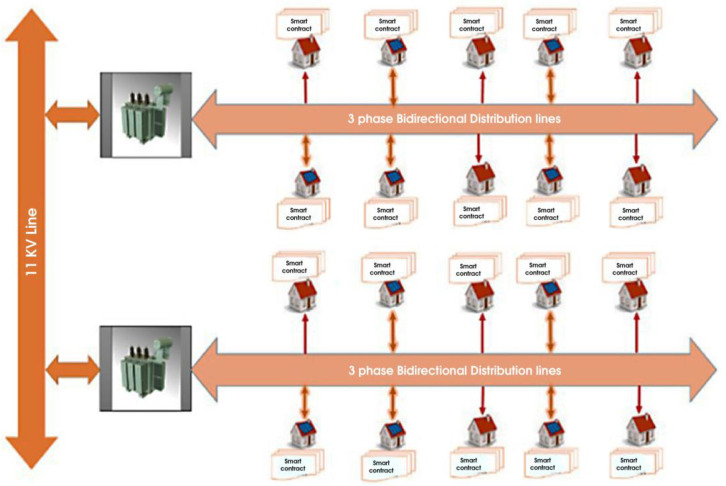
Blockchain-enabled smart-grid distribution network model.

**Figure 5 sensors-24-01668-f005:**
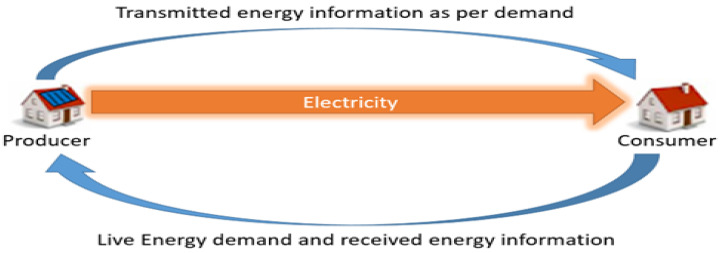
Smart-contract energy trading.

**Figure 6 sensors-24-01668-f006:**
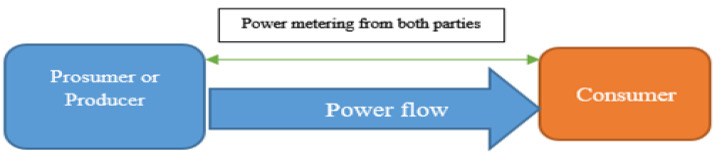
Smart-contract energy transaction P2P.

**Figure 7 sensors-24-01668-f007:**
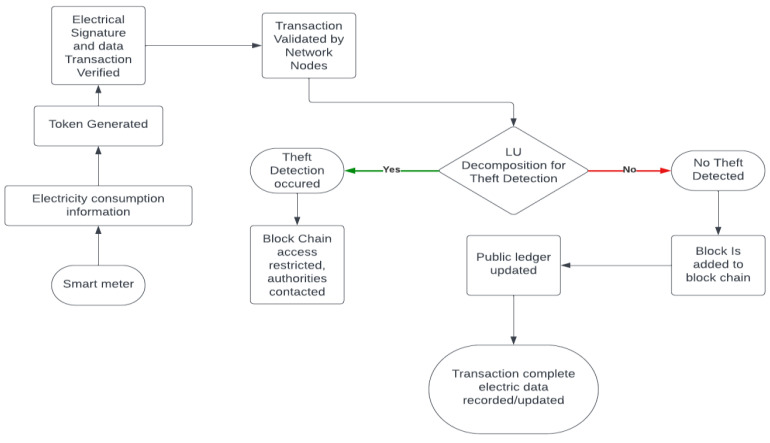
Flow chart of recording data for electricity transactions.

**Figure 8 sensors-24-01668-f008:**
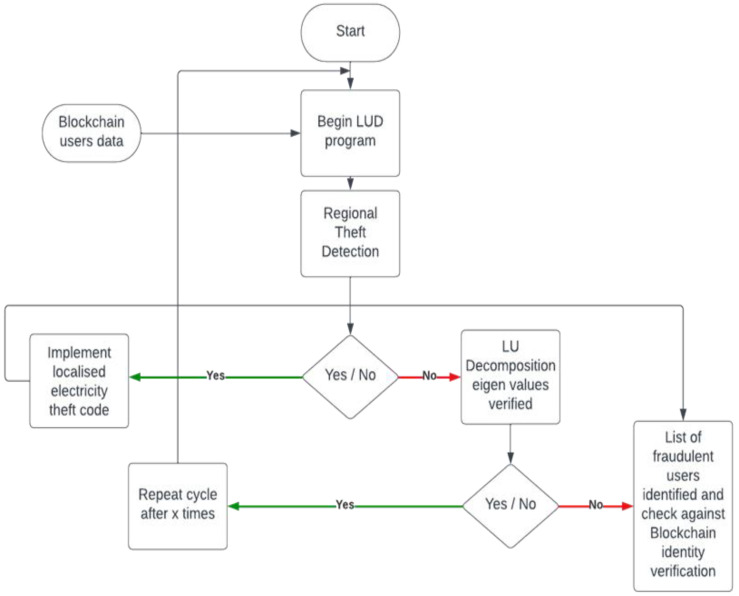
Working flowchart of OLUD algorithm.

**Figure 9 sensors-24-01668-f009:**
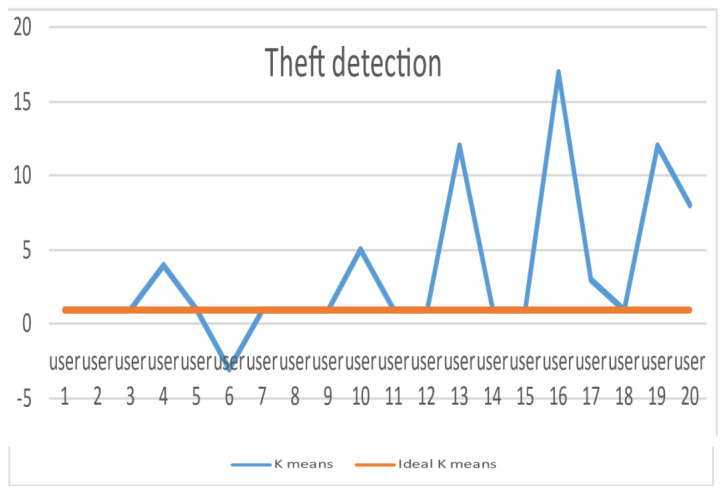
Elements of k for ideal-system versus real-world scenario.

**Figure 10 sensors-24-01668-f010:**
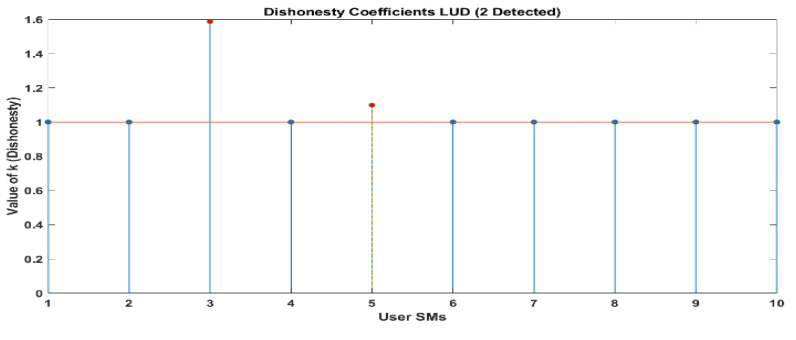
Dishonesty coefficient for LUD, 10 users.

**Figure 11 sensors-24-01668-f011:**
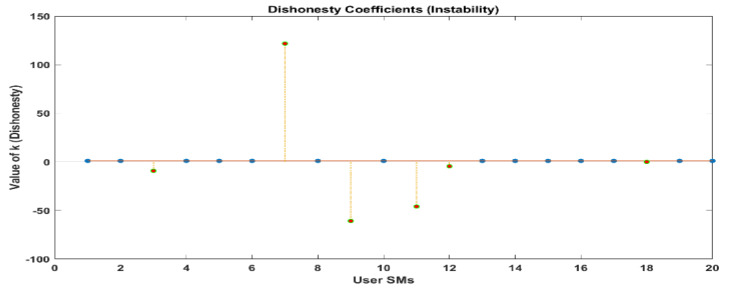
Dishonesty coefficient for LUD (instability), 20 users.

**Figure 12 sensors-24-01668-f012:**
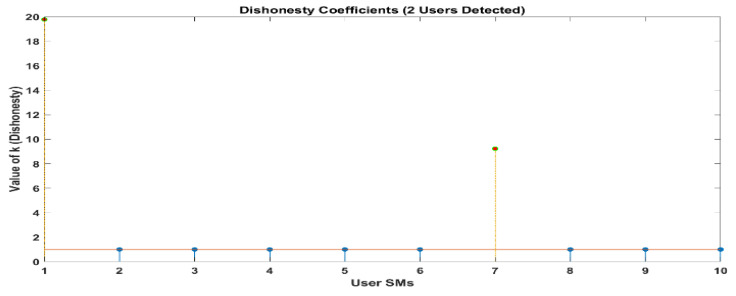
Dishonesty coefficient for LUDP, 10 users.

**Figure 13 sensors-24-01668-f013:**
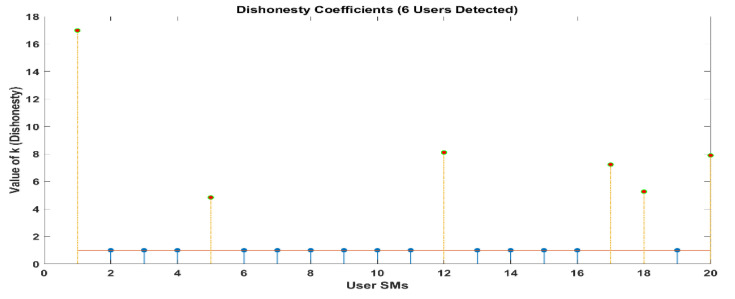
Dishonesty coefficient for LUDP (stability), 20 users.

**Figure 14 sensors-24-01668-f014:**
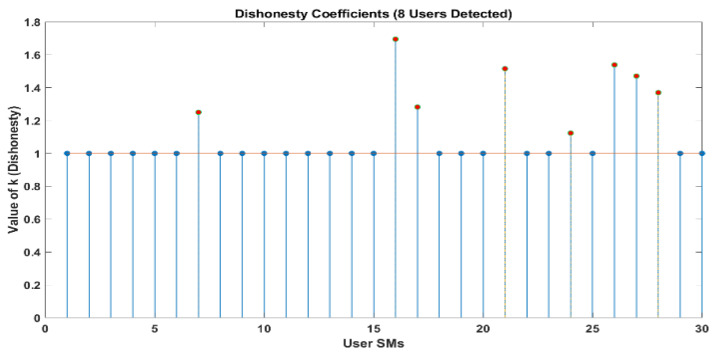
Elements of k obtained in a network of 20 users using the OLUD algorithms.

**Figure 15 sensors-24-01668-f015:**
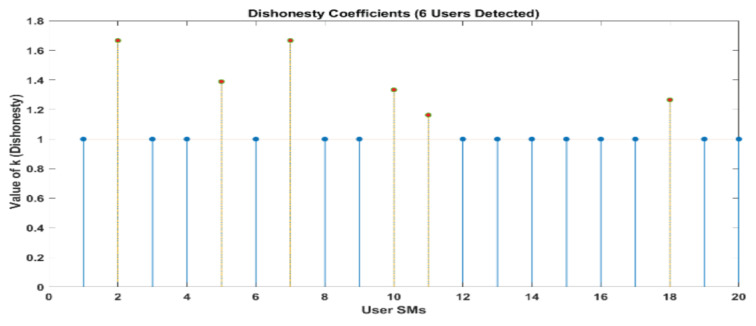
Elements of k obtained in a network of 30 users using the OLUD algorithm.

**Figure 16 sensors-24-01668-f016:**
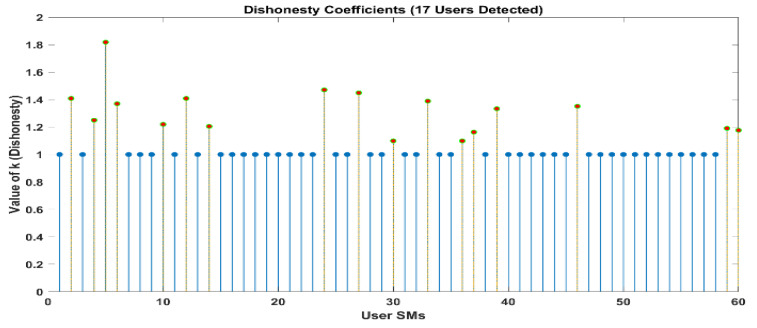
Elements of k obtained in a network of 60 users using the OLUD algorithms.

**Figure 17 sensors-24-01668-f017:**
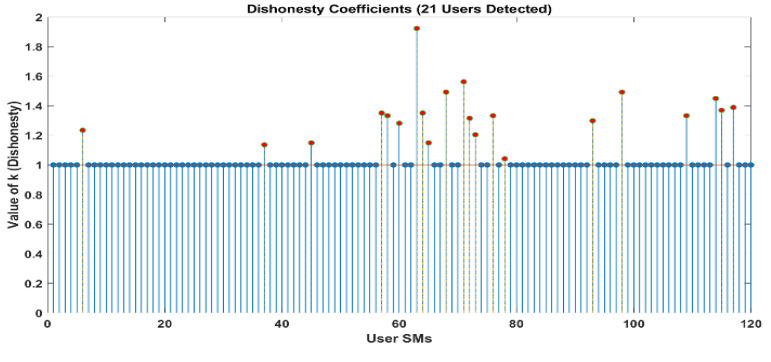
Elements of k obtained in a network of 120 users using the OLUD algorithm.

**Table 1 sensors-24-01668-t001:** Comparison of proposed technique and previous studies [l1].

Paper	Technique Adopted	Attack	Defense	Key Findings
[30]	Deep neural network model	Cyber attacks	Deep neural network model	High accuracy in classifying cyber-attacks in smart grids
[31]	Data aggregation	Data manipulation	Consortium blockchain	Improved performance for multidimensional data acquisition
[32]	Privacy-preserving data aggregation	User’s identity	Multiple pseudonyms, private blockchains	Improved user privacy in power grid communications
[33]	FedDetect, deep learning model	Energy-theft detection	Secure protocol, deep learning model	Secure and efficient energy-theft detection
[34]	CNN-LSTM- based model	Electricity-theft identification	Data from smart meters	Good accuracy in identifying fraudulent actions
[35]	LSTM-based evaluation method	System-stability condition determination	Evaluation method using LSTM	Better performance than other established evaluation techniques
[36]	Combined DL- boosting model	NTL detection	Feature extraction using DL-boosting model	Improved detection of electricity theft
Proposed	Optimized LUD, integrating blockchain	Various attacks on AMI	Privacy-preserving, theft detection	Theft detection with more accuracy and better performance than other established evaluation techniques

**Table 2 sensors-24-01668-t002:** Comparison between LUD, LUDP, and OLUD based on stability, network size and complexity.

Method	Stability	Network Size	Complexity
LUD	no	fewer than 20	simple
LUDP	yes	any	complex
OLUD	yes	any	simple

**Table 3 sensors-24-01668-t003:** Elements of k for ideal-system versus real-world scenario.

User No.	K Means	Ideal K Means
1	1	1
2	1	1
3	1	1
4	4	1
5	1	1
6	−3	1
7	1	1
8	1	1
9	1	1
10	5	1
11	1	1
12	1	1
13	12	1
14	1	1
15	1	1
16	17	1
17	3	1
18	1	1
19	12	1
20	8	1

## Data Availability

All processed data used in this study are included in the article.

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
