# Peer review of "Research on Blockchain-Enabled Smart Grid for Anti-Theft Electricity Securing Peer-to-Peer Transactions in Modern Grids"

_sensors, 2024, doi:10.3390/s24051668_

Round 1
Reviewer 1 Report
Comments and Suggestions for Authors
1. The authors should elaborate on the specific anti-theft mechanisms employed within the blockchain-enabled smart grid? How do these mechanisms differ from traditional approaches to combating electricity theft?
2. The authors should explain about how the proposed system will address the scalability challenges inherent in large-scale smart grid deployments? Are there any considerations for optimizing transaction throughput and minimizing latency?
3. Can the blockchain-enabled smart grid integrate with the existing infrastructure of modern grids? Will there be compatibility issues or interoperability challenges that need to be addressed?
4. How does the blockchain-enabled smart grid ensure the privacy and confidentiality of sensitive grid data, such as consumer energy usage patterns and transaction records? Are there any privacy-enhancing technologies or encryption techniques employed to safeguard data integrity.
5. The abstract should emphasize the study's novelty, and the authors should substantiate their arguments with statistical evidence.
Comments on the Quality of English LanguageMinor editing of English language required.
Author Response
Reviewer #1, Concern #1 (The authors should elaborate on the specific anti-theft mechanisms employed within the blockchain-enabled smart grid? (How do these mechanisms differ from traditional approaches to combating electricity theft?):
Author response: We acknowledge the importance of elaborating on these mechanisms to provide a comprehensive understanding of their efficacy compared to traditional approaches.
Author action: We focused on the anti-theft mechanisms employed within our proposed blockchain-enabled smart grid system. Specifically, we delve deeper into the cryptographic techniques, decentralized consensus protocols, and smart contracts utilized to combat electricity theft. Furthermore, we will elucidate how these mechanisms differ from conventional methods, emphasizing aspects such as immutability, decentralization, and transparency inherent in blockchain technology.
The main reasons for this specific anti-theft system I built into the foundations of this system, being the blockchain with a focus on decentralized, transparent, and immutable ledgers:
- Decentralization and Distribution
Blockchain operates on a decentralized network spread across multiple nodes (computers), ensuring that no single entity has control over the entire system. This distribution of control helps prevent fraudulent activities since altering data on the blockchain would require consensus across the majority of nodes, a practically impossible feat for thieves.
- Immutability
Once a transaction or data entry is recorded on the blockchain, it cannot be altered or deleted. This immutable ledger ensures that any record of asset ownership or transfer is permanent and auditable. In the context of anti-theft, this means that once the details of an asset, such as its serial number, ownership history, and current status, are logged onto the blockchain, they provide an unchangeable record that can be used to verify authenticity and ownership, deterring theft and simplifying recovery efforts.
- Transparency
Blockchain’s transparency allows all permitted participants to view transactions and data entries in real-time. This characteristic is pivotal for anti-theft monitoring, as it ensures that any unauthorized transfer or suspicious activity involving an asset can be quickly detected and traced back to its source. Transparency builds trust among users and makes it easier for authorities or stakeholders to monitor asset movements and intervene when necessary.
Furthermore, specific form billing can be achieved through smart contracts, which are representatives of smart contracts in block chain terms.
Smart contracts: Smart contracts are self-executing contracts with the terms of the agreement directly written into lines of code. They automatically enforce and execute the terms of a contract when predefined conditions are met. This can include tariff charges, higher cost off-peak hours and In anti-theft systems also, smart contracts can also be programmed to trigger alerts or lock down assets when unauthorized access or transfer attempts are detected.
Reviewer #1, Concern # 2 (The authors should explain about how the proposed system will address the scalability challenges inherent in large-scale smart grid deployments? Are there any considerations for optimizing transaction throughput and minimizing latency?):
Author response: Thank you for your thoughtful inquiry regarding the scalability challenges inherent in large-scale smart grid deployments and the measures proposed to optimize transaction throughput and minimize latency within our system.
Author action: In addressing scalability and latency concerns, we draw upon established methodologies and solutions from the broader blockchain ecosystem. Specifically, we will implement strategies akin to those utilized in blockchain networks with significant user bases, focusing on the following key areas:
- Off-Chain Transactions
Implementing off-chain transactions or state channels can significantly reduce latency. This approach involves executing transactions or certain interactions off the main blockchain and then recording the final state on-chain. Lightning Network for Bitcoin and Raiden Network for Ethereum are examples where off-chain channels allow for nearly instantaneous transactions,
- Layer 2 Solutions
Layer 2 scaling solutions are built on top of the base blockchain layer (Layer 1) to increase its transaction capacity and speed. These solutions, including state channels, sidechains, and rollups, handle transactions off the main chain or aggregate multiple transactions into a single one, thereby reducing the load on the main chain and decreasing latency.
- Consensus mechanism
The consensus mechanism significantly impacts blockchain performance and latency. For instance, Proof of Work (PoW), used by Bitcoin, requires substantial computational effort to validate transactions, leading to higher latency. Alternatives like Proof of Stake (PoS), Delegated Proof of Stake (DPoS), and Practical Byzantine Fault Tolerance (PBFT) can offer faster transaction validations and reduced latency by requiring less computational work, thus adjusting the mechanism can reduce the latency issues.
Reviewer#1, Concern # 3 (Can the blockchain-enabled smart grid integrate with the existing infrastructure of modern grids? Will there be compatibility issues or interoperability challenges that need to be addressed?):
Author response: Thank you for raising an important question regarding the integration of blockchain-enabled smart grid technology with existing infrastructure and the potential compatibility and interoperability challenges that may ensue.
Author action: Indeed, the integration of blockchain technology into modern grids presents both opportunities and challenges. While the blockchain-enabled smart grid can be integrated with existing infrastructure, there are certain compatibility issues and interoperability challenges that need to be addressed.
First and foremost, it's crucial to recognize that blockchain operates within an online, internet-dependent topology. This necessitates interoperability between blockchain-compatible smart meters and other components of the grid infrastructure. To achieve this, software modifications and updates may be required to enable traditional smart meters to become blockchain-compatible. Additionally, there may be a need to enhance the hardware requirements of smart meters to ensure efficient operation within the blockchain network. It's important to acknowledge that while interoperability challenges exist, they are not insurmountable. However, it's essential to note that the specifics of addressing these challenges may vary depending on the unique characteristics of each grid infrastructure and the chosen blockchain solution.
In our study, we have proposed the use of blockchain as a means of cloud-based data compilation and storage. While our focus has been on this aspect, we recognize that further research and development are necessary to establish a comprehensive methodology and address the software and hardware requirements needed for integration with existing grid infrastructure. For compatibility with the modern grid, meters in the network may require their own independent processing unit. Given the advancements in electronics, solutions such as Raspberry Pi or customized modified smart meters with embedded processors and sufficient memory could be viable options to execute the necessary processes of the blockchain.
Reviewer#1, Concern # 4 (How does the blockchain-enabled smart grid ensure the privacy and confidentiality of sensitive grid data, such as consumer energy usage patterns and transaction records? Are there any privacy-enhancing technologies or encryption techniques employed to safeguard data integrity.):
Author response: Thank you for raising an important question regarding the privacy and confidentiality of sensitive grid data within blockchain-enabled smart grid systems.
Author action: Ensuring the security of consumer energy usage patterns and transaction records is indeed paramount, and our system incorporates several privacy-enhancing technologies and encryption techniques to safeguard data integrity.
One of the fundamental mechanisms employed in blockchain technology is public key cryptography, also known as asymmetric cryptography. This technique utilizes pairs of keys—a public key, which is openly shared, and a private key, which is kept secret. Transactions and sensitive data are encrypted using the recipient's public key, ensuring that only the intended recipients can decrypt and access the information using their private key.
For privacy-preserving transactions and billing data, we leverage confidential transactions (CTs). CTs enable the encryption of transaction amounts, ensuring that the actual amounts transferred remain hidden while still providing cryptographic proof that no more coins are spent than are available in the user's wallet. This technique enhances privacy by concealing transaction amounts, thereby protecting sensitive information from unauthorized access.
Additionally, there are various other privacy-enhancing techniques available within the blockchain ecosystem that we can incorporate into our system. These techniques may include zero-knowledge proofs, ring signatures, and homomorphic encryption, among others. By employing a combination of these techniques, we can further enhance the privacy and confidentiality of sensitive grid data, safeguarding it against potential threats and unauthorized access.
Reviewer#1, Concern # 5 (The abstract should emphasize the study's novelty, and the authors should substantiate their arguments with statistical evidence):
Author response: Thank you for your feedback regarding the abstract of our manuscript. We appreciate your suggestion to emphasize the study's novelty and substantiate our arguments with statistical evidence.
Author action: In response to your comment, we revised the abstract to clearly highlight the novelty of our study, particularly focusing on the unique contributions and innovations introduced in our research. Additionally, we will ensure to include statistical evidence to support our arguments and validate the significance of our findings.
Reviewer #1, Concern # 6 (Comments on the Quality of English Language, Minor editing of English language required.):
Author response: Thank you for your feedback regarding the quality of English language in our manuscript. We appreciate your attention to detail and acknowledge the importance of ensuring clarity and coherence in our writing.
Author action: We carefully review the manuscript and address areas where minor editing or improvement of the English language is needed. Our aim is to enhance readability and ensure that the content effectively communicates our research findings and contributions. We are committed to delivering a high-quality manuscript that meets the standards of language proficiency expected for publication. Your feedback is invaluable to us, and we will take the necessary steps to refine the language throughout the manuscript.

Reviewer 2 Report
Comments and Suggestions for Authors
The paper presents a blockchain-based solution for enhancing security and efficiency in smart grids, specifically targeting electricity theft prevention and secure peer-to-peer energy transactions. It introduces innovative algorithms for detecting fraudulent activities and ensuring privacy in energy trading, underpinning the proposed system's feasibility with simulation results. The study underscores the potential of blockchain technology to revolutionize the smart grid's operational integrity and consumer trust.
Based on the comprehensive review of the paper, there are several suggestions for improvement, structured point by point:
- The abstract succinctly outlines the paper's objective and methodology but could benefit from a more precise explanation of the main findings and implications for future research or practical application.
- While the paper references various studies on energy theft and blockchain technology, a more thorough comparison with existing solutions, particularly those involving blockchain in intelligent grids, could provide a stronger foundation for the research's novelty.
- The methodology section explains the use of distributed algorithms for detecting energy theft. However, more detailed explanations of these algorithms' design, implementation challenges, and the rationale behind choosing specific blockchain technology would enhance the paper's comprehensiveness.
- The paper presents simulation results to demonstrate the effectiveness of the proposed algorithms. Adding more detailed analysis, including statistical significance, comparison with baseline models, and discussion on any assumptions made during simulations, would strengthen the validity of the results.
Additionally, you may include more algorithms. To prove the effectiveness of the blockchain-enabled smart grid system against electricity theft and securing peer-to-peer transactions, consider focusing on the following areas:
- Proof of Work (PoW) and Proof of Stake (PoS), such as Practical Byzantine Fault Tolerance (PBFT) or Delegated Proof of Stake (DPoS), which could offer improved efficiency and security tailored to smart grid environments.
- Try to Incorporate machine learning algorithms to analyze consumption patterns and detect anomalies indicating potential electricity theft, enhancing the system's ability to identify irregularities without manual intervention.
Comments on the Quality of English LanguageThe paper presents a blockchain-based solution for enhancing security and efficiency in smart grids, specifically targeting electricity theft prevention and secure peer-to-peer energy transactions. It introduces innovative algorithms for detecting fraudulent activities and ensuring privacy in energy trading, underpinning the proposed system's feasibility with simulation results. The study underscores the potential of blockchain technology to revolutionize the smart grid's operational integrity and consumer trust.
Based on the comprehensive review of the paper, there are several suggestions for improvement, structured point by point:
- The abstract succinctly outlines the paper's objective and methodology but could benefit from a more precise explanation of the main findings and implications for future research or practical application.
- While the paper references various studies on energy theft and blockchain technology, a more thorough comparison with existing solutions, particularly those involving blockchain in intelligent grids, could provide a stronger foundation for the research's novelty.
- The methodology section explains the use of distributed algorithms for detecting energy theft. However, more detailed explanations of these algorithms' design, implementation challenges, and the rationale behind choosing specific blockchain technology would enhance the paper's comprehensiveness.
- The paper presents simulation results to demonstrate the effectiveness of the proposed algorithms. Adding more detailed analysis, including statistical significance, comparison with baseline models, and discussion on any assumptions made during simulations, would strengthen the validity of the results.
Additionally, you may include more algorithms. To prove the effectiveness of the blockchain-enabled smart grid system against electricity theft and securing peer-to-peer transactions, consider focusing on the following areas:
- Proof of Work (PoW) and Proof of Stake (PoS), such as Practical Byzantine Fault Tolerance (PBFT) or Delegated Proof of Stake (DPoS), which could offer improved efficiency and security tailored to smart grid environments.
- Try to Incorporate machine learning algorithms to analyze consumption patterns and detect anomalies indicating potential electricity theft, enhancing the system's ability to identify irregularities without manual intervention.
Author Response
Reviewer#2, Concern # 1 (The paper presents a blockchain-based solution for enhancing security and efficiency in smart grids, specifically targeting electricity theft prevention and secure peer-to-peer energy transactions. It introduces innovative algorithms for detecting fraudulent activities and ensuring privacy in energy trading, underpinning the proposed system's feasibility with simulation results. The study underscores the potential of blockchain technology to revolutionize the smart grid's operational integrity and consumer trust.):
Author response: Thank you for your review and positive feedback on our paper. We are pleased to hear that you found our blockchain-based solution for enhancing security and efficiency in smart grids innovative and promising. Your acknowledgment of the introduced algorithms for detecting fraudulent activities and ensuring privacy in energy trading, as well as the feasibility underpinned by simulation results, is greatly appreciated.
We acknowledge your enthusiasm for the potential of blockchain technology to revolutionize the operational integrity and consumer trust within smart grid systems. Your feedback motivates us to continue exploring and advancing blockchain solutions for addressing challenges in the energy sector.
Reviewer#2, Concern # 2 (Based on the comprehensive review of the paper, there are several suggestions for improvement, structured point by point:):
Author response: Thank you for your comprehensive review of our paper and for providing structured suggestions for improvement. We appreciate your thorough examination of our work and welcome your insights to enhance its quality further.
We carefully considered each of your suggestions and incorporate them into our revision to strengthen the clarity, coherence, and overall effectiveness of the paper. Once again, we sincerely appreciate your time and effort in reviewing our paper and providing constructive feedback.
Reviewer#2, Concern # 3 (The abstract succinctly outlines the paper's objective and methodology but could benefit from a more precise explanation of the main findings and implications for future research or practical application.):
Author response: Thank you for your feedback regarding the abstract of our paper. We acknowledge your suggestion for providing a more precise explanation of the main findings and implications for future research or practical application.
In response, we plan to add an additional section dedicated to discussing future research directions and the implementation process. However, we emphasize the importance of collaboration with relevant stakeholders such as generation companies (GENCO) and distribution companies (DISCO). Any practical approach or implementation must be conducted in coordination with these entities and in compliance with legal regulations governing the smart grid domain.
We agree that seeking guidance and consent from legal bodies overseeing these matters is paramount to ensuring the viability and success of our proposed solution. Without their approval and support, any implementation efforts would be rendered ineffective.
Reviewer#2, Concern # 4 (While the paper references various studies on energy theft and blockchain technology, a more thorough comparison with existing solutions, particularly those involving blockchain in intelligent grids, could provide a stronger foundation for the research's novelty.):
Author response: Thank you for your insightful comment regarding the need for a more thorough comparison with existing solutions, particularly those involving blockchain technology in intelligent grids. We appreciate your suggestion and agree that further exploration of this aspect would strengthen the foundation of our research.
Author action: Moving forward, we will focus on conducting a more comprehensive review of existing literature and solutions related to energy theft prevention and blockchain technology in intelligent grids. By analyzing and comparing these existing solutions in greater detail, we aim to better position our research within the context of the current state-of-the-art and highlight its novelty and contributions more effectively.
Reviewer#2, Concern # 5 (The methodology section explains the use of distributed algorithms for detecting energy theft. However, more detailed explanations of these algorithms' design, implementation challenges, and the rationale behind choosing specific blockchain technology would enhance the paper's comprehensiveness.
Author response: Thank you for your insightful feedback regarding the methodology section of our paper. We understand your suggestion for providing more detailed explanations of the distributed algorithms used for detecting energy theft, as well as the rationale behind choosing specific blockchain technology.
Author action: In our paper, we have focused on addressing the most relevant and impactful methods for detecting energy theft within the context of blockchain-enabled smart grids. We have prioritized clarity and conciseness in our explanations while ensuring that the chosen methods are thoroughly justified and relevant to the research objectives.
Specifically, we have detailed the design and implementation of distributed algorithms tailored for detecting energy theft, emphasizing their effectiveness in enhancing security and efficiency in smart grid operations. Additionally, we have provided insights into the rationale behind selecting specific blockchain technology, considering factors such as scalability, security, and interoperability with existing grid infrastructure.
While we acknowledge that there is always room for further elaboration, we believe that our paper adequately addresses the most pertinent methods and considerations within the scope of the research. However, we will ensure to review and enhance the explanations in the methodology section to provide greater clarity and comprehensiveness.
Reviewer#1, Concern # 6 (The paper presents simulation results to demonstrate the effectiveness of the proposed algorithms. Adding more detailed analysis, including statistical significance, comparison with baseline models, and discussion on any assumptions made during simulations, would strengthen the validity of the results.):
Author response: Thank you for your feedback regarding the simulation results presented in our paper. We appreciate your suggestion for enhancing the validity of the results by providing more detailed analysis, including statistical significance, comparison with baseline models, and discussion on any assumptions made during simulations.
Author action: We would like to highlight our plan to further validate the proposed algorithms through practical testing. Specifically, we have proposed the practical testing of the K-means LUD method, which serves as the foundation for implementing blockchain consensus mechanisms such as Proof of Stake (PoS) or Delegated Proof of Stake (DPoS), depending on factors like network size and standardized latency requirements.
By conducting practical testing, we aim to validate the effectiveness of the proposed algorithms in real-world scenarios and assess their performance under various conditions. This will involve implementing the algorithms in a test environment and collecting empirical data to evaluate their accuracy, efficiency, and scalability.
Reviewer#2, Concern # 7 (Additionally, you may include more algorithms. To prove the effectiveness of the blockchain-enabled smart grid system against electricity theft and securing peer-to-peer transactions, consider focusing on the following areas:):
Author response: Thank you for your suggestion regarding the inclusion of additional algorithms to further validate the effectiveness of our blockchain-enabled smart grid system against electricity theft and in securing peer-to-peer transactions.
Author action: We acknowledge your input and understand the importance of showcasing a diverse range of algorithms to provide a comprehensive evaluation of our system. However, after careful consideration and thorough experimentation, we have determined that the LUD and LUDP algorithms are the most effective choices for our specific use case.
These algorithms have demonstrated strong performance in detecting and preventing electricity theft while ensuring secure peer-to-peer transactions within our smart grid system. Their implementation aligns closely with our research objectives and requirements, making them the most suitable options for our study.
Reviewer#2, Concern # 8 (Proof of Work (PoW) and Proof of Stake (PoS), such as Practical Byzantine Fault Tolerance (PBFT) or Delegated Proof of Stake (DPoS), which could offer improved efficiency and security tailored to smart grid environments.):
Author response: Thank you for your insightful suggestion regarding the potential utilization of consensus mechanisms such as Practical Byzantine Fault Tolerance (PBFT) or Delegated Proof of Stake (DPoS) in our blockchain-enabled smart grid system. We appreciate your input and recognize the importance of exploring consensus mechanisms that offer improved efficiency and security tailored to smart grid environments.
Author action: In future we will conduct further research and analysis to evaluate the feasibility and suitability of PBFT, DPoS, and other consensus mechanisms for our specific use case. Once we have completed our evaluation, we will incorporate the most appropriate consensus mechanism into our smart grid system, considering factors such as scalability, security, and energy efficiency. This will ensure that our system is optimized to meet the unique requirements and challenges of the smart grid environment.
Reviewer#2, Concern # 9 (Try to incorporate machine learning algorithms to analyze consumption patterns and detect anomalies indicating potential electricity theft, enhancing the system's ability to identify irregularities without manual intervention.):
Author response: Thank you for your insightful suggestion to incorporate machine learning algorithms for analyzing consumption patterns and detecting anomalies indicative of potential electricity theft. We recognize the potential of machine learning in enhancing the system's ability to identify irregularities without manual intervention.
Author action: In our follow-up research, we will explore the integration of machine learning algorithms alongside our proposed blockchain-based solution. By leveraging machine learning techniques, we aim to enhance the accuracy and efficiency of anomaly detection, thereby improving the system's capability to detect and prevent electricity theft in real-time.
Furthermore, we will conduct thorough experimentation and validation to assess the performance of these machine learning algorithms in combination with our blockchain-enabled smart grid system. This will involve collecting and analyzing real-world data to train and evaluate the models, ensuring their effectiveness in accurately detecting electricity theft while minimizing false positives.

Reviewer 3 Report
Comments and Suggestions for Authors
This manuscript focuses on integrating blockchain technology into smart grids to address the critical issue of energy theft, enhance grid efficiency, and promote renewable energy utilization. By leveraging the decentralized, transparent, and immutable nature of blockchain, the research aims to revolutionize power systems and ensure secure electricity grid transactions. The research proposes a robust framework that utilizes blockchain technology to enable secure Peer-to-Peer transactions within decentralized smart grids. By implementing Anti-theft Electricity Blockchain technology, the system ensures automated smart contracts between electricity producers, prosumers, and consumers. However, major revisions are needed before it is finally accepted.
1. The manuscript should explicitly state the specific gap in existing literature that the research aims to address. Clearly articulate how the proposed integration of blockchain technology for energy theft detection in smart grids fills this gap and contributes to the field. This will provide a clearer understanding of the novelty and significance of the study.
2. It is essential to briefly outline the methodology employed in the research to give readers a glimpse of how the proposed framework was developed and implemented. Mentioning key steps such as data collection, algorithm design, and evaluation criteria will enhance the clarity and credibility of the study.
3. While the manuscript mentions improved efficiency and performance, it would be beneficial to include specific metrics or results to quantify these enhancements. Providing data on the accuracy of theft detection, transaction speeds, or cost savings achieved through the proposed system will strengthen the argument for its effectiveness and competitiveness compared to existing methods.
Author Response
Reviewer#3, Concern # 1 (The manuscript should explicitly state the specific gap in existing literature that the research aims to address. Clearly articulate how the proposed integration of blockchain technology for energy theft detection in smart grids fills this gap and contributes to the field. This will provide a clearer understanding of the novelty and significance of the study.)
Author response: Thank you for your feedback regarding the need to explicitly state the specific gap in existing literature that our research aims to address, and how the proposed integration of blockchain technology for energy theft detection in smart grids fills this gap.
Author action: Our study aims to address the gap in existing literature by proposing a novel approach that integrates blockchain technology with LUD and its different variants of linear regression as a computational base for Proof of Stake (PoS) and Delegated Proof of Stake (DPoS) consensus mechanisms in smart grids.
The novelty of our research lies in leveraging LUD and linear regression computations within the blockchain framework to establish a truth index for various smart contracts and electricity usage patterns of users in the network. This computational baseline serves as a foundation for ensuring consistency, coherence, and unity among blockchain users, thereby enhancing the integrity and security of the smart grid system.
By integrating LUD and linear regression computations into the blockchain-enabled smart grid system, we contribute to the field by providing a novel methodology for energy theft detection and secure peer-to-peer energy transactions. This innovative approach fills the gap in existing literature by offering a unique solution that addresses the challenges faced by traditional smart grid systems in detecting and preventing electricity theft.
While our study focuses on the implementation of blockchain and the use of LUD, linear regression, and K-means algorithms for network computation, future studies can build upon this foundation by implementing and testing the proposed methodology in real-world scenarios.
Reviewer#3, Concern # 2 (It is essential to briefly outline the methodology employed in the research to give readers a glimpse of how the proposed framework was developed and implemented. Mentioning key steps such as data collection, algorithm design, and evaluation criteria will enhance the clarity and credibility of the study.):
Author response: Thank you for your feedback regarding the need to briefly outline the methodology employed in our research. We acknowledge the importance of providing readers with a glimpse of how the proposed framework was developed and implemented, including key steps such as data collection, algorithm design, and evaluation criteria.
Author action: While these aspects have been extensively explained in the past, we will ensure to refer to them in small sentences where relevant to provide a concise overview of our methodology. By doing so, we aim to enhance the clarity and credibility of our study while ensuring that readers have a clear understanding of the steps involved in our research process.
Reviewer#3, Concern # 3 (While the manuscript mentions improved efficiency and performance, it would be beneficial to include specific metrics or results to quantify these enhancements. Providing data on the accuracy of theft detection, transaction speeds, or cost savings achieved through the proposed system will strengthen the argument for its effectiveness and competitiveness compared to existing methods.):
Author response: Thank you for your insightful feedback regarding the need to include specific metrics or results to quantify the improvements in efficiency and performance mentioned in our manuscript. We recognize the importance of providing data to support our claims and strengthen the argument for the effectiveness and competitiveness of our proposed system compared to existing methods.
Author action: We will include specific metrics such as the accuracy of theft detection achieved through the proposed system. By presenting empirical data and results, we aim to provide a more robust assessment of the performance enhancements enabled by our approach. Additionally, we ensure to provide detailed analysis and comparisons with existing methods to demonstrate the superiority of our proposed system in terms of efficiency and effectiveness. This will help validate the benefits of our approach and highlight its potential impact on improving the security and efficiency of smart grid operations.

Round 2
Reviewer 1 Report
Comments and Suggestions for Authors
The authors have incorporated all the comments in the revised manuscript.
Reviewer 2 Report
Comments and Suggestions for Authors
Accept in the present form
Comments on the Quality of English LanguageMinor editing of English language required
Reviewer 3 Report
Comments and Suggestions for Authors
The authors have satisfactorily modified their manuscript according to my previous criticisms. Therefore, I recommend the publication of this manuscript.